# Smart Agriculture and Rural Revitalization and Development Based on the Internet of Things under the Background of Big Data

**Xi Ma**

School of Foreign Language, Yuxi Normal University, Yuxi 653100, China; daisyma@yxnu.edu.cn

**Abstract:** Smart agriculture refers to the specific performance of the smart economy in the field of agriculture; it is a form of agricultural smart economy and an important part of the smart economy. It has played a certain role in promoting rural revitalization and development. The purpose of this paper was to study the role of smart agriculture based on the Internet of Things in rural revitalization and development under the background of big data. The purpose was to use Internet of Things technology to realize smart agriculture under the background of big data, so as to promote rapid rural revitalization and development. Therefore, in this paper, a fuzzy PID algorithm and genetic algorithm were proposed. Finally, through experimental analysis, the fuzzy PID algorithm was used to carry out experiments in the laboratory. The temperature and humidity of the laboratory were measured. The average difference between the collected and actual temperature values was 0.6 °C, and the maximum difference between the collected and actual humidity values was 1.32% RH. The laboratory simulation results satisfied the performance indicators and technical requirements of the system. The system operated normally and could be directly applied to field tests. The experimental results show that the role of Internet of Things technology in the smart agricultural economy is irreplaceable, which further illustrates the positive relationship between smart agriculture based on the Internet of Things and rural revitalization and development. As one of the most mature technologies in today's society, the Internet of Things technology combined with smart agriculture not only offers new perspectives, but also promotes the revitalization and development of rural areas, indicating a new direction for its future research.

**Keywords:** smart agriculture; rural revitalization; Internet of Things technology; big data

## 1. Introduction

IoT technology is an important breakthrough point of informatization. In recent years, many new technologies have appeared in agricultural planting and production. The agricultural Internet of Things comprises the application of technologies such as "digitization, networking, and intelligence" in agricultural production. It provides certain supports for agricultural production and can play an important role in promoting rural revitalization and development. Smart agriculture is a new trend in the development of agricultural science and technology, which is the comprehensive application of a series of high-end technologies such as computer technology and network communication technology. Intelligent agriculture refers to the introduction of the Internet of Things technology into agricultural production, and its system functions include: monitoring, real-time imaging and video monitoring. It provides help for the rational use of agricultural resources, reduces production costs, improves economic efficiency, improves the ecological environment, and promotes rural revitalization. This paper used a neural network algorithm and fuzzy PID algorithm to study the role of IoT technology in smart agriculture. General trends in the evolution of research methods and recent advances were examined to advance rural revitalization.

The innovations of this paper are as follows: (1) Regarding the research discussion on smart agriculture, according to the current literature, there is a lack of similar research involving data analysis. The induction of future research and development trends in smart agriculture can be realized. (2) The design of a fuzzy PID algorithm was targeted. The values of temperature and humidity in an agricultural production environment were quantitatively analyzed, and the limitations of remote monitoring of agricultural information were comprehensively considered. (3) Using a genetic algorithm, agricultural production and agricultural soil parameters were accurately calculated.

## 2. Related Work

At present, there are many academic research studies on smart agriculture and rural revitalization. Ren-Wei H E (2022) believed that one of the most important ways of upgrading capital for farmers was to improve agricultural production efficiency and promote rural economic development. Social capital in the form of social resources or social networks is one of the most important forms of livelihood capital for farmers, which can improve agricultural production efficiency and increase farmers' incomes [1]. Yang J (2021) proposed to use remote sensing technology, positioning technology and other counting methods to analyze the development of the rural economy and study agricultural productivity from the perspective of rural tourism [2]. Benessaiah K (2021) believed that the return of farmers to their hometowns was conducive to promoting the development of the rural economy, which could also further improve the agricultural productivity of the countryside [3]. Yang R (2021) proposed the idea of regional differences and drivers of rural vulnerability. It strengthened the prediction and monitoring of disturbance sources of rural revitalization and development, thereby improving rural agricultural production capacity and promoting the rural revitalization [4]. Zhang X (2022) proposed some strategies for agricultural English translation that could provide reference for future academic research on agricultural English translation [5]. However, the above studies were analyses of rural revitalization and development, and there is a lack of research on the combination of smart agriculture and rural revitalization. Therefore, a scientific method is urgently needed for verification.

At present, there are many research studies on the Internet of Things technology in academia. Wei D (2021) believed that the proliferation of smart IoT allowed electronic sensors and devices to connect and generate streaming data streams, and put forward a test for its efficient management [6]. Ferrag M A (2022) proposed the use of technologies such as IoT and 5G/6G communications to manage COVID-19 and other epidemics. In this study, a comprehensive understanding and analysis of the related technologies of the Internet of Things and its imaging field were carried out [7]. Kouimtzis T H (2021) proposed to use IoT technology for medical treatment. In root reconstruction, previously treated immature maxillary central incisors were treated with IoT technology [8]. Oubenali N's (2022) research aimed to use IoT technology to interpret and analyze medical concepts in the field of visualization using word embedding methods [9]. Refaee E A (2022) believed that technologies such as artificial intelligence, IoT, blockchain and deep learning had revolutionized the development of many fields, especially the medical field. It could use these technologies to timely and accurately diagnose patients' conditions [10]. The above studies have fully analyzed the Internet of Things technology, but they have not been combined with the study of rural economic development. The source of references to relevant works in the article is Baidu Academic.

## 3. Methods of IoT-Based Smart Agriculture

*3.1. Development of Smart Agriculture Using Internet of Things Technology*

The Internet of Things technology effectively combines sensors and embedded technology to achieve information exchange between networks and develop comprehensive technologies such as intelligent identification, data analysis and processing in the interaction process [11]. The IoT structure is divided into three layers, namely the application

layer, network layer, and perception layer. Different layers implement different functions. The specific architecture is shown in Figure 1.

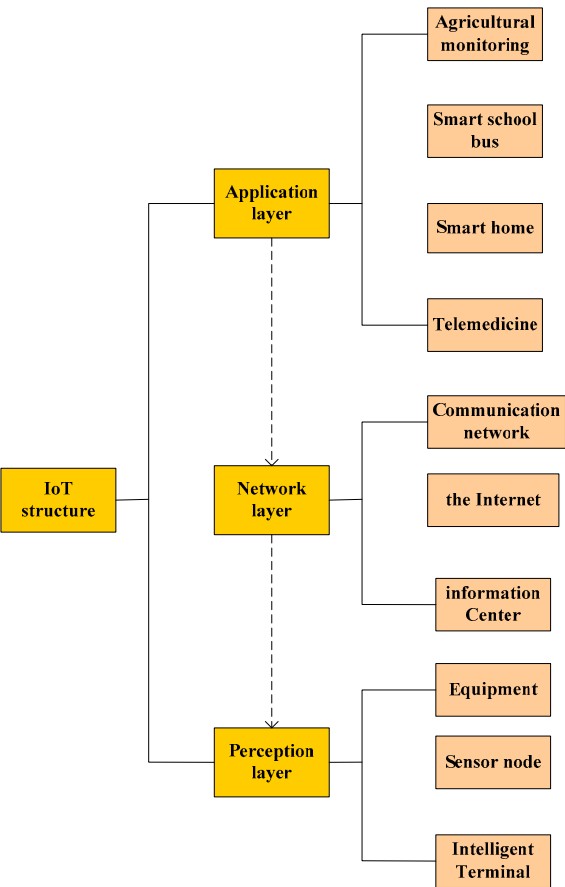

**Figure 1.** IoT architecture.

Smart agriculture refers to the combination of science and technology and agricultural production methods to achieve intelligent, automated and unmanned management processes [12]. This indicates that agricultural production has entered a new stage. It integrates a number of new technologies such as the Internet of Things and the mobile Internet, which help make the agricultural production environment intelligent and conducive to promoting the development of rural revitalization [13]. Shown in Figure 2 is a diagram of scenarios related to smart agriculture.

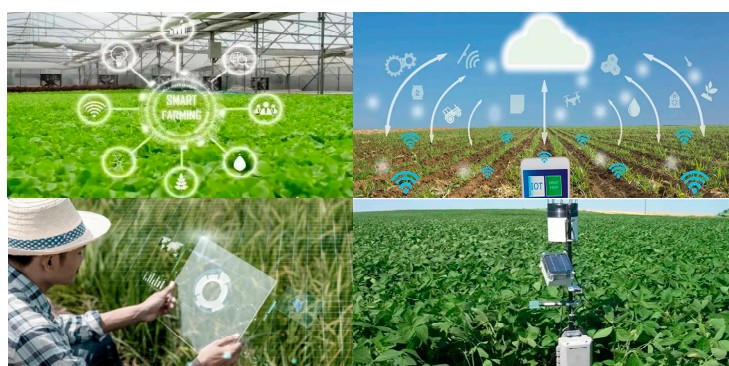

**Figure 2.** Scenarios related to smart agriculture.

This paper used a fuzzy PID algorithm and genetic algorithm to analyze smart agriculture.

### 3.2. Fuzzy PID Algorithm

The source for the formula pictures was a research study on a variable-rate granular fertilizer control system based on a fuzzy PID algorithm.

In simulation experiments, fuzzy PID control is the method most commonly used for the controller, as shown in Figure 3.

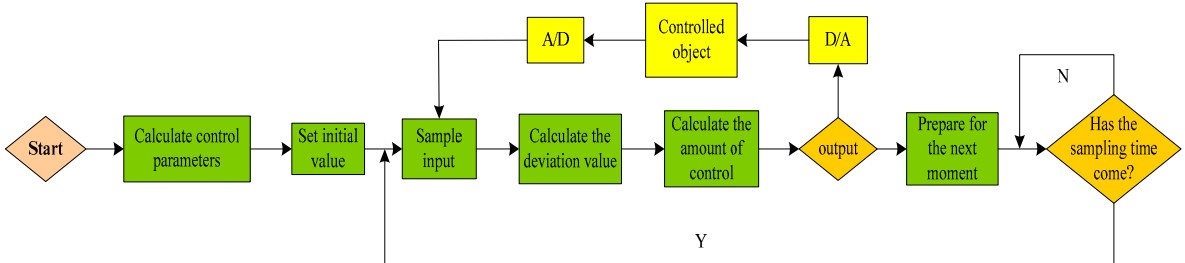

**Figure 3.** Block diagram of an analog PID control system.

It can be seen from Figure 3 that the PID controller is a linear controller that needs to generate a control deviation between the determined value and the actual output value [14]:

$$A(t) = v(t) - o(t) \tag{1}$$

where $v(t)$ represents the fixed value, and $o(t)$ represents the actual output value.

The control form of the PID controller is shown in Formula (2):

$$u(t) = K_q \left[ A(t) + \frac{1}{T_i} \int_0^t A(t)d(t) + T_d \frac{dA(t)}{dt} \right] \tag{2}$$

It can also take the form of its Ctrip transfer function as:

$$F(s) = \frac{V(s)}{E(S)} = K_q \left( 1 + \frac{1}{T_I s} + T_D s \right) \tag{3}$$

where $K_q$ is the proportional coefficient, $T_I$ is the integral time constant, and $T_D$ is the differential time constant.

When the increase in the control amount becomes the variable of the executor, then:

$$u(k) = K_q e(k) + K_I \sum_{j=0}^{k} e(j) + K_D [e(k) - e(k-1)] \tag{4}$$

The PID control formula is calculated. According to the recursive form, the following formula can be obtained:

$$u(k-1) = K_q e(k-1) + K_I \sum_{j=0}^{k-1} e(j) + K_D [e(k-1) - e(k-2)] \tag{5}$$

Formulas (4) and (5) are subtracted to obtain:

$$\Delta u(k) = K_q [e(k) - e(k-1)] + K_I e(k) + K_D [e(k-1) - e(k-2)] = K_q \Delta e(k) + K_D [\Delta e(k) - \Delta e(k-1)] \tag{6}$$

where

$$\Delta e(k) = e(k) - e(k-1) \tag{7}$$

Therefore, the PID control algorithm is expressed by Formula (6).

Formula (6) can be further rewritten as:

$$\Delta u(k) = Xe(k) - Ye(k-1) + Ze(k-2) \tag{8}$$

where

$$X = K_q(1 + \frac{T}{T_I} + \frac{T_D}{T}) \tag{9}$$

$$Y = K_q(1 + 2\frac{T_D}{T}) \tag{10}$$

$$Z = K_q T_D / T \tag{11}$$

To sum up, when the sampling period $T$ of the computer control system is constant, then only the values of $K_q$, $K_I$, and $K_D$ are required, and the control increment can be obtained by Formulas (6) and (7) [15]. When the incremental algorithm is used, the control increment output by the computer corresponds to the increment of this power [16]. As shown in Figure 4, it is the program diagram of the incremental PID control algorithm.

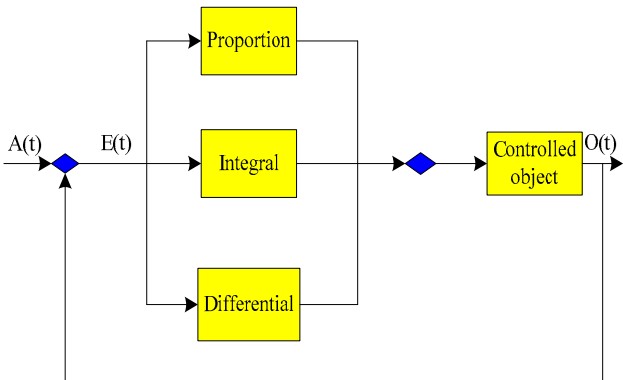

**Figure 4.** Program diagram of incremental PID control algorithm.

The PID control algorithm was used to adjust the PID control parameters of an agricultural production variable fertilization control system [17], as shown in Tables 1 and 2.

**Table 1.** Relationship between PID tuning parameters and system time domain performance indicators.

| Parameter | $K_q$ | $K_I$ | $K_D$ |
|---|---|---|---|
| Rise time | −1 | −1 | 0 |
| Excessive process time | 0 | 1 | −1 |
| Overshoot | 1 | 1 | −1 |
| Static error | −1 | Eliminate | 0 |

**Table 2.** Critical proportionality tuning controller parameters.

| Controller Type | P | PI | PID |
|---|---|---|---|
| Scale $\eta/\%$ | $3\eta_k$ | $2.3\eta_k$ | $1.8\eta_k$ |
| Integration time $T_i$ | $\infty$ | $0.824T_k$ | $0.51T_k$ |
| Differential time $T_d$ | 0 | 0 | $0.123T_k$ |

Among them, −1 indicates that the value is decreasing, 1 indicates that the value is increasing, and 0 indicates that the number has not changed significantly. Comparing the parameters of the PID controller according to the critical proportional method, the proportional value is determined as $\eta_k = 9.37\%$, and the oscillation period is $T_k = 1.6s$ according to the parameters of the response curve of the object when the critical oscillation occurs.

It can be seen from Table 2 that the tuning parameters of the control algorithm are $K_Q = 6.33$, $K_D = 7.3$, and $K_I = 4.13$, respectively.

### 3.3. Genetic Algorithms

A genetic algorithm is an algorithm used to imitate the process of biological evolution. In past studies, it has been shown that it is difficult to obtain the global optimal value by a simple genetic algorithm [18]. In order to ensure the global convergence of the genetic algorithm, some scholars have proposed an adaptive genetic algorithm. Next, some adaptive genetic algorithms were briefly introduced [19]. Some scholars have proposed an improved genetic algorithm, which is an adaptive genetic algorithm for crossover probability and mutation probability. The expressions of these two probabilities are as follows:

$$P_o = \begin{cases} \frac{k_1(g_{max}-g')}{g_{max}-g_{avg}} & g' \geq g_{avg} \\ k_2 & g' < g_{avg} \end{cases} \tag{12}$$

$$P_x = \begin{cases} \frac{k_3(g_{max}-g)}{g_{max}-g_{avg}} & g \geq g_{avg} \\ k_4 & g < g_{avg} \end{cases} \tag{13}$$

where $k_1, k_2, k_3, k_4 \in (0,1)$ and $g_{max}$ represent the largest fitness values in the population, $g_{avg}$ represents the average fitness value of each generation of the population, $g'$ is the larger fitness value of the two crossover individuals, and $g$ represents the fitness value of the mutant individual.

On the basis of the adaptive genetic algorithm, a new algorithm, the improved adaptive genetic algorithm, was developed [20]. Its crossover probability and mutation probability are expressed as:

$$P_o = \begin{cases} P_{o1} - \frac{(P_{o1}-P_{o2})(g'-g_{avg})}{g_{max}-g_{avg}} & g' \geq g_{avg} \\ P_{o1} & g' < g_{avg} \end{cases} \tag{14}$$

$$P_x = \begin{cases} P_{x1} - \frac{(P_{x1}-P_{x2})(g_{max}-g')}{g_{max}-g_{avg}} & g' \geq g_{avg} \\ P_{x1} & g' < g_{avg} \end{cases} \tag{15}$$

where $P_{o1}, P_{o2}, P_{x1}, P_{x2} \in (0,1)$ can be adjusted during the optimization process. $g_{max}$ represents the largest fitness value in the population, and $g_{avg}$ represents the average fitness value of each generation of the population.

## 4. Experiments in Smart Agriculture Based on IoT Technology

### 4.1. System Test Using PID Algorithm for Smart Agriculture Experiment

The performance of the system hardware equipment in the simulated use process in the laboratory environment was as follows:

(1) Wireless transmission distance: The main factors affecting the wireless transmission distance were the transmission method, transmission content and occlusion. When there was no obstruction between the data acquisition module and the controller, the transmission distance was 70 m, and the transmission distance between the controllers was 2 km. Wireless network communication was used for transmission between the controller and the server. It was not limited by distance, but was affected when the network signal coverage strength was insufficient due to factors such as occlusion.

(2) Environmental data collection: The data collection situation was simulated and verified in the laboratory. Through specific data collection and real-time monitoring results analysis, the data results were normal, and there was no abnormal data situation. The simulated environmental monitoring effect was in line with the expected situation and the law of crop growth demand. However, due to improper operation of the equipment or limitations of the service life of the equipment, the equipment experienced hardware failures and other phenomena.

(3) Experimental safety and real-time performance: The experimental scheme was designed with full consideration for safety factors, and corresponding safety precautions for the system were taken within the context of the safety scheme. Real-time mainly referred to the timely processing of data to ensure that the data was received in a timely manner during the transmission process.

(4) Experimental environment: laboratory of Agricultural University. The main experimental instruments were as follows:

1) Temperature measuring instrument: measured the temperature change during the experiment.

The high-precision multichannel thermometer is a multi-channel electrical signal and temperature collector, and can also be used for temperature calibration.

2) Humidity meter: measured the humidity change during the experiment.

The HDC2022 is an integrated humidity and temperature sensor that uses an ultra-compact WLCSP (wafer level, chip level package) to provide high-precision measurements with ultra-low power consumption.

3) Wireless transmission distance measuring instrument: transmitted and stored experimental data.

4) Environmental data acquisition instrument: detected the environment before, during and after the experiment.

(5) Experimental process: since the growth of crops is closely related to the surrounding environment, this paper mainly performed statistics and analysis of the temperature and humidity changes under the two systems.

*4.2. Results of System Testing Using PID Algorithm for Smart Agriculture*

(1) Laboratory accuracy test: The accuracy of the data collected by the sensor directly affected the transmission performance of the system and the stability of the application. Therefore, in the laboratory environment, the requirements for air temperature and humidity were high, and high accuracy of data values was required to ensure the efficient operation of the system. Therefore, the system tested and verified the accuracy of air temperature and humidity readings in the laboratory environment.

This experiment was conducted in the laboratory of China Agricultural University. The system was connected to a temperature and humidity sensor after commissioning, and the indoor temperature and humidity were used as test objects. The actual indoor temperature and humidity were the standards for calculating the test accuracy. The temperature and humidity values were collected five times in the test, and the specific data are shown in Tables 3 and 4.

**Table 3.** Temperature Experimental Data.

| Temperature | 1 | 2 | 3 | 4 | 5 |
|---|---|---|---|---|---|
| Measured value | 20.27 | 21.29 | 21.30 | 20.98 | 21.53 |
| Actual value | 21.21 | 21.47 | 21.97 | 20.45 | 20.87 |
| Error value | 0.88 | 0.17 | 0.67 | 0.53 | 0.69 |

**Table 4.** Humidity Experimental Data.

| Humidity | 1 | 2 | 3 | 4 | 5 |
|---|---|---|---|---|---|
| Measured value | 29.51 | 30.09 | 30.11 | 30.45 | 29.96 |
| Actual value | 30.33 | 30.52 | 29.17 | 31.14 | 30.28 |
| Error value | 0.38 | 0.12 | 0.40 | 0.37 | 0.29 |

It can be seen from Table 3 that the difference between the collected value and the actual value of temperature is within 1 °C, indicating that the system performance of the laboratory is very stable.

It can be seen from Table 4 that the difference between the collected value and the actual value of humidity is within 0.5% RH, which further indicates that the system of the laboratory operates normally and is very stable.

According to Tables 3 and 4, the average difference between the collected and actual temperature values is 0.6 °C, and the maximum difference between the collected and actual humidity values is 1.32% RH. The laboratory simulation results can satisfy the performance indicators and technical requirements of the system. The system operates normally and can be directly applied to the field test.

(2) Field test: The field test was carried out in a rural greenhouse to simulate the experiment. The on-site control system and computer were used for hardware equipment control, network communication, and platform monitoring function tests. The sensor equipment in the field was installed in a control cabinet.

The system operation results showed that the field experimental equipment could accurately and efficiently collect environmental data in the greenhouse, and the monitoring platform could display the indoor parameter values online in real time. The electric button of the control cabinet could realize the control of the on-site system, and could also realize the switch between manual and automatic, which could effectively control the normal operation of the supporting equipment. The computer was used for data transmission. The software platform realized real-time online monitoring, data storage and control, and all parameter settings were preset.

In the laboratory, the stability and reliability of the operating state of the system were verified, and the system achieved accurate measurements and introduced the fuzzy PID algorithm to adjust the accuracy of the sensor. In order to accurately monitor the change in data on greenhouse temperature, the data storage and processing cycle positioning was 1m. The control system performed a parameter determination every 12 s. When the fuzzy controller performed one control, the fuzzy decision-making program was executed once every 45 s, and each humidification was executed for 45 s. Therefore, when performing the fine operation, the fuzzy decision-making program was executed once every 30 s, each humidification operation lasted for 10 s, and the state of the decision-making process remained unchanged. When the controller implemented decision-making, the comparison between using the fuzzy PID compound algorithm and not using it was as shown in Figure 5.

Figure 5a shows the temperature test values with and without the PID algorithm in the experimental group. In the process of using the fuzzy PID algorithm to control fine adjustment, the temperature deviation after stabilization was within 0.2 °C. During the fine-tuning process without fuzzy PID control, the temperature error after stability was within 0.5 °C. Figure 5b shows the humidity test values of the experimental group with and without the PID algorithm. When the algorithm was used for humidity measurement, the humidity deviation was within 2.5% RH. When this method was not used to measure greenhouse humidity, the appropriate deviation was within 3% RH. Compared with traditional control methods, the accuracy of temperature and humidity control was significantly improved, which indicated that this temperature and humidity satisfied the requirements of crop growth.

## 4.3. Results of Smart Agriculture Using Genetic Algorithms

In order to test the improved performance of the network, a comparison test of classification and recognition was carried out. The comparison targets were the traditional SFAM network and the ssFAM, aaGAM, and ssEAM networks. The data characteristics selected above are shown in Table 5.

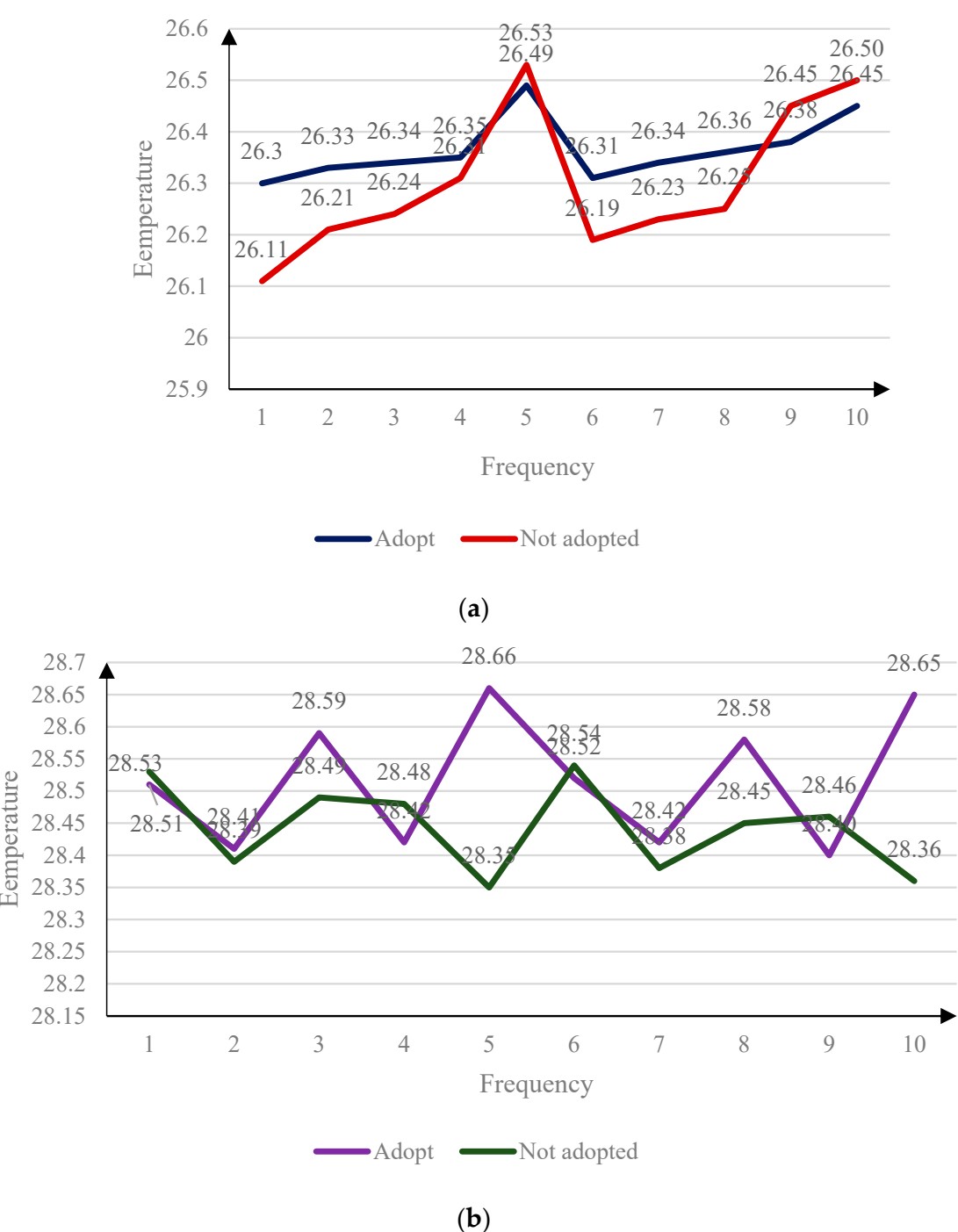

**Figure 5.** Temperature and humidity test values for the experimental group. (**a**) Temperature test values for experimental group with and without PID algorithm. (**b**) Humidity test values for experimental group with and without PID algorithm.

The parameters of the initial network in the first stage: vigilance parameter $\theta_a = 0$, network weight $W_j^a = 1$, learning efficiency $\alpha = 1.2$, $\beta = 0.002$. For the genetic algorithm, the initial network population size was set to $Pop_{size} = 21$, the maximum algebra is $Gen_{\max} = 101$, and the gene string in the population was set to be a random sequence.

**Table 5.** Test Database Feature Description.

| Name Database | Number of Database Samples | Training Pattern Library | Forensic Pattern Library |
|---|---|---|---|
| Iris | 160 | 80 | 70 |
| Wine | 180 | 80 | 80 |
| Glass | 215 | 101 | 101 |
| Name database | Test pattern library | Category | Feature amount |
| Iris | 80 | 4 | 5 |
| Wine | 103 | 4 | 14 |
| Glass | 115 | 5 | 11 |

The initial population of the genetic algorithm in the second stage is set to the network weights obtained from the training mode determined in the first stage. The way to determine the alert parameters in the network is:

$$\theta_a^{step} = \frac{\theta_a^{\max} - \theta_a^{\min}}{\theta_{size} - 1} \tag{16}$$

$$\theta_a^i = \theta_a^{\min} + i \times \theta_a^{step}, i = 0, 1, 2, \ldots, pop_{size} - 1 \tag{17}$$

where $\theta_a^i$ represents the vigilance parameter of the *i*-th network. It is supposed that $\theta_a^{\max} = 0.96$, $\theta_a^{\min} = 0.11$, and the network parameter is $\alpha = 0.2, \beta = 0.002$.

According to the above experiments, five experiments were carried out under the same experimental conditions, and the average was obtained. The experimental results are shown in Table 6.

**Table 6.** Comparison of Experimental Results.

| Name Database | MAGASFAM | | ssFAM | | SFAM | | ssEAM | | ssGAM | |
|---|---|---|---|---|---|---|---|---|---|---|
| | Accuracy | Number of Class Nodes | Accuracy | Number of Class Nodes | Accuracy | Number of Class Nodes | Accuracy | Number of Class Nodes | Accuracy | Number of Class Nodes |
| Iris | 92.95 | 2 | 90.26 | 3 | 92.10 | 2 | 92.24 | 2 | 93.3 | 2 |
| Wine | 90.23 | 5 | 88.27 | 6 | 90.55 | 4 | 89.78 | 4 | 90.47 | 4 |
| Glass | 81.24 | 7 | 76.66 | 9 | 79.77 | 7 | 79.25 | 7 | 80.11 | 6 |

It can be seen from Table 6 that during the experiment, the comparison was made in terms of the experimental accuracy and the number of class nodes in the experiment. From the experimental data, it was found that the Iris accuracy of the MAGASFAM part was 92.95, and the number of class nodes was 2. The accuracy rate of the Wine part reached 90.23, and the number of class nodes was 5. The Glass part had an accuracy of 81.24, with 7 class nodes. The MAGASFAM network obtained by improving the network learning rules by the genetic algorithm had obvious differences and improvements from the traditional SFAM, ssFAM, ssGAM and ssEAM networks in terms of data accuracy and number of class nodes. Through the above experiments, it could be predicted that the complexity of features and the scale of sample points to be classified were increasing. Compared with the traditional SFAM, ssFAM, ssGAM, and ssEAM networks, the performance of the MAGASFAM network could be greatly improved in terms of data accuracy and number of class nodes.

However, in the process of the experiment, it was also found that the time spent by the MAGASFAM network in the experimental process was significantly longer than that of the SFAM, ssFAM, ssGAM, and ssEAM network experiments. This was mainly because the MAGASFAM network obtained its weights through the genetic algorithm, resulting in the improved MAGASFAM network spending more time in experimental training and learning. Of course, in the process of using the genetic algorithm to process the network

in stages, in order to reduce the training time as much as possible without impairing the accuracy of the network, a simple genetic algorithm was used for experimental learning in the first stage. However, with the continuous advancement of science and technology, the performance of computer networks has also been constantly improving, and the difficulties in experiments have also been overcome. It can be seen that the performance and training structure of the MAGASFAM network were improved according to the training structure and performance of the SFAM, ssFAM, ssGAM, and ssEAM networks, which were improved to a certain extent.

*4.4. Improved Experimental Scheme Using Fuzzy PID Algorithm and Genetic Algorithm*

Agricultural greenhouses, as one of the most effective methods in agricultural planting, can be used as research sites to avoid the influence of other factors and make the experimental data more accurate. This chapter mainly used a fuzzy PID algorithm and genetic algorithm for experimental testing. The PID algorithm was used to measure the temperature and humidity in the agricultural greenhouse, and the genetic algorithm was used to analyze various nutrient indicators of the soil in agricultural planting. In the process of analysis, the advantages of various algorithms were fully utilized. However, there were still some defects in the experimental process, the details of which are as follows. When experimenting with the PID fuzzy algorithm, only one location was tested for humidity and temperature. It was not possible to form experimental control data, and there was no reference basis, which was very unfavorable for understanding the accuracy of the experimental results. Therefore, it was necessary to add a control group and test the temperature and humidity at another site to form an experimental comparison, which was conducive to analyzing whether the various data of the experimental group met the actual requirements. When using the genetic algorithm to analyze soil composition, a variety of network learning models were used for testing. However, the time-consuming process of the experiment was much slower than that of the traditional network algorithm, which was inconvenient for the subsequent experimental analysis. It was necessary to improve the genetic algorithm or find a better optimal alternative algorithm to solve the problem of the algorithm taking too long to process. As shown in Figure 6, the data analysis diagram of the control group was added under the fuzzy PID algorithm.

Compared with Figure 5a, Figure 6a shows the temperature test values for the control group with and without the PID algorithm. After the experiment was stable, the temperature deviation of the data was within 0.6 °C, which was 0.4 °C higher than the corresponding 0.2 °C deviation for the experimental group. In the fine-tuning process without using fuzzy PID algorithm control, the temperature deviation after stabilization was within 0.9 °C, which was higher than the 0.5 °C deviation for the experimental group. In the humidity measurement process, compared with Figure 5b, Figure 6b shows the humidity test values for the control group with and without the PID algorithm. Its humidity deviation was within 4% RH, 1.5% higher than the 2.5% relative humidity deviation for the experimental group. When the greenhouse humidity was not measured by this method, the humidity deviation was within 4% relative humidity, which was 1% higher than the 3% relative humidity deviation for the test group. Compared with the experimental group, the temperature and humidity values for the control group showed an upward trend, indicating that the crops in the control group area could be planted with heat-resistant varieties.

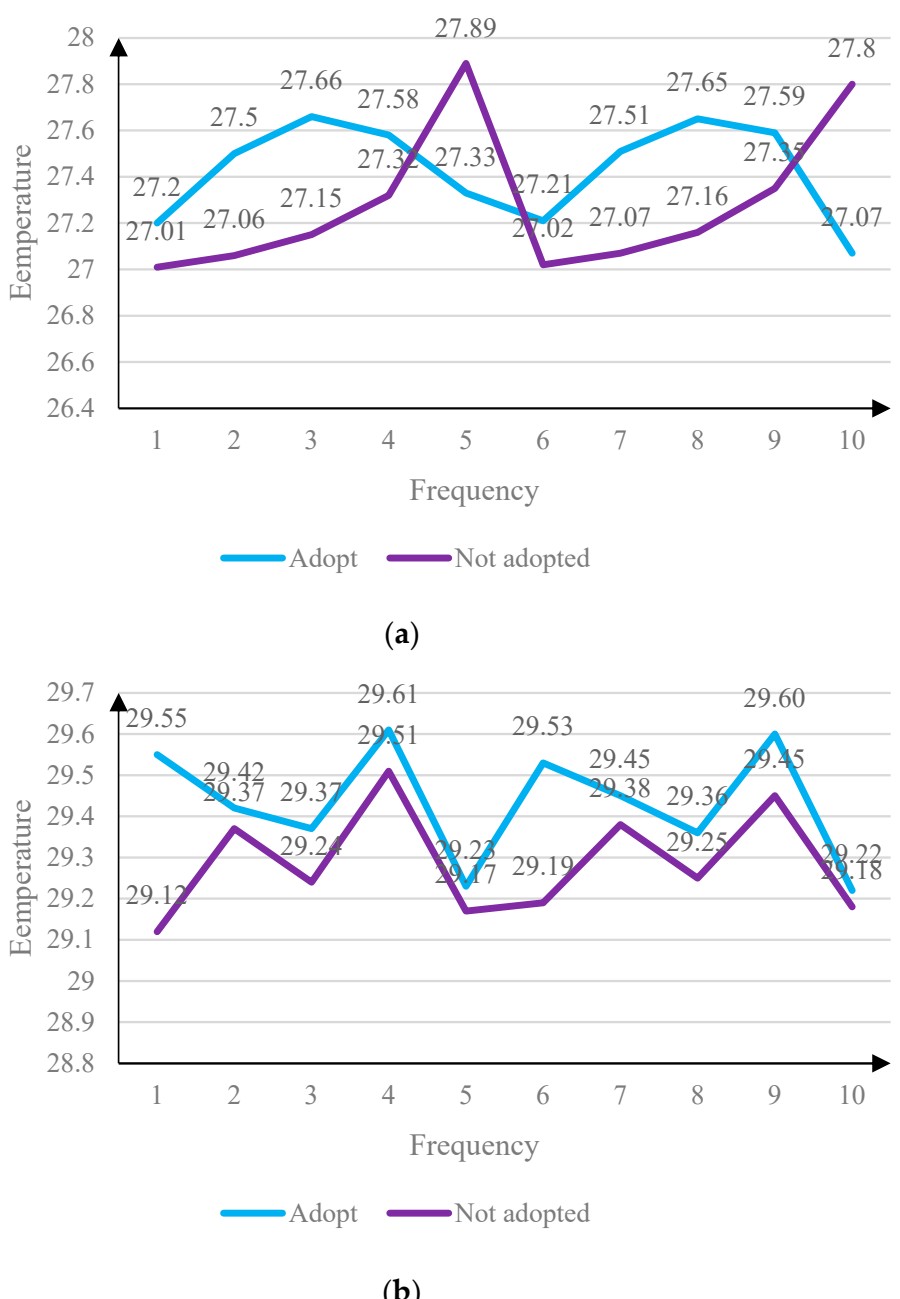

**Figure 6.** Temperature and humidity test values for the control group. (**a**) Temperature test values for control group with and without PID algorithm. (**b**) Humidity test values for control group with and without PID algorithm.

## 5. Discussion

This paper mainly used a fuzzy PID algorithm and genetic algorithm to analyze the effect of smart agriculture on rural revitalization and development. It first described the analysis method used. Then, in the experiment, the system hardware performance index was analyzed, and it was indicated that the experiment could be carried out only when the performance index met the test conditions. It also discussed the security and timeliness of the system, which can ensure its safe and stable operation. The experimental data were analyzed and compared in two environments: laboratory and greenhouse. Through laboratory experiments, it was proved that the system could be operated under greenhouse conditions and that each index met the requirements of experiments under

greenhouse conditions. Under the greenhouse conditions, the fuzzy PID algorithm was used for verification, showing that the algorithm could effectively improve and control the accuracy. The genetic algorithm was used to analyze the differences between the MAGASFAM network and other networks in the analysis of soil indicators. Additionally, it was concluded that the MAGASFAM network required more time in the experiment than the other networks.

In the context of big data, this paper used the Internet of Things system to analyze the development of smart agriculture and rural revitalization. There are not many studies on the use of IoT to analyze agricultural development in academia. The research in this paper aimed to further expand and extend the application scope of Internet of Things technology and to analyze the research on smart agriculture and rural revitalization and development from another angle. The above experiments demonstrated the effectiveness of the fuzzy PID algorithm and genetic algorithm in smart agriculture analysis.

## 6. Conclusions

In this paper, within the context of the Internet of Things, a fuzzy PID algorithm and genetic algorithm were used to analyze the temperature, humidity and soil nutrient composition of an agricultural production environment. Through the above analysis, the following results were obtained. The fuzzy PID algorithm was used to analyze the temperature and humidity of an agricultural production environment. Different temperature-tolerant plants should be grown under different temperature and humidity conditions. With the genetic algorithm, the MAGASFAM network pair took a long time in the experiment and still had certain defects. In summary, more experimental analysis is needed in the study of smart agriculture and rural revitalization and development. These need to be considered from many aspects and comprehensively analyzed in order to promote the development of the rural economy. This paper found that in the context of big data, smart agriculture based on the Internet of Things is not only an important direction for agricultural development in the future, but can also greatly improve the revitalization and development of rural areas, improve the development of the rural economy, and promote the process of urban–rural integration.

**Funding:** This research received no external funding.

**Institutional Review Board Statement:** Not applicable.

**Informed Consent Statement:** Not applicable.

**Data Availability Statement:** Not applicable.

**Conflicts of Interest:** The author declares no conflict of interest.

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
