# Peer review of "Smart Agriculture and Rural Revitalization and Development Based on the Internet of Things under the Background of Big Data"

_sustainability, doi:10.3390/su15043352_

Round 1

Reviewer 1 Report

The publication does not meet the formal requirements. Tables should be reduced in size, for example: tables 1, 2. Tables 3, 4 are collapsed. Unfortunately Figures 5 and 6 are so schematic that they show almost no difference. The parts of the tables (a and b) are almost the same.

Additionla comments: I find the publication too general. Much more literature would be better if it were included. Data collection was done 5 times. For me, this is very few measurements. You should support your conclusions with more data. The schematic diagrams you have produced tell us nothing. My suggestion: take more measurements.

Author Response

Comments and Suggestions for Authors

The publication does not meet the formal requirements. Tables should be reduced in size, for example: tables 1, 2. Tables 3, 4 are collapsed. Unfortunately Figures 5 and 6 are so schematic that they show almost no difference. The parts of the tables (a and b) are almost the same.

Answer: I have modified all the tables in the article and deleted Figures 5 and 6.

Additionla comments: I find the publication too general. Much more literature would be better if it were included. Data collection was done 5 times. For me, this is very few measurements. You should support your conclusions with more data. The schematic diagrams you have produced tell us nothing. My suggestion: take more measurements.

Answer:I have changed the number of times to collect data to 10, and modified the original picture.

Reviewer 2 Report

In its current form, the article adds no new content and cannot be published.

The article contains many editorial and typing errors. These include missing spaces between elements of the article, lack of periods after ordinal numbers lack of citation of sources of figures and formulas, poor formatting of text in tables, etc. The noted errors were marked in color in the text of the article and commented on and sent in the appendix to this review. The article lacks a properly described test methodology and a listing of the measuring instruments and equipment used during the tests, as well as a test plan. The work should also be stylistically rewritten so that there are not, for example, several consecutive sentences starting identically, and a clear formulation of the purpose, scope and course of the research.

In such a version, the work cannot be published. It should be corrected and supplemented. The author did not discuss his own research results with those of other researchers and did not present clear conclusions. In the discussion, he only included an extended summary of his achievements.

Given the number of such errors caught, the entire text should be rewritten and checked for editorial correctness again.

Also, its correctness in terms of English language is questionable to me, so it should be checked by a native-speaker who is familiar with this field of agricultural technology, information technology and knows the relevant technical vocabulary.

Author Response

Comments and Suggestions for Authors

In its current form, the article adds no new content and cannot be published.

The article contains many editorial and typing errors. These include missing spaces between elements of the article, lack of periods after ordinal numbers lack of citation of sources of figures and formulas, poor formatting of text in tables, etc. The noted errors were marked in color in the text of the article and commented on and sent in the appendix to this review. The article lacks a properly described test methodology and a listing of the measuring instruments and equipment used during the tests, as well as a test plan. The work should also be stylistically rewritten so that there are not, for example, several consecutive sentences starting identically, and a clear formulation of the purpose, scope and course of the research.

Answer:The source of references in relevant work in the article is Baidu Academic; The source of the formula pictures is Research on variable rate granular fertilizer control system based on fuzzy pid algorithm,Experimental environment: the laboratory of an agricultural university. The main experimental instruments are: temperature measuring instrument and humidity measuring instrument, wireless transmission distance measuring instrument, and environmental data collection instrument.

In such a version, the work cannot be published. It should be corrected and supplemented. The author did not discuss his own research results with those of other researchers and did not present clear conclusions. In the discussion, he only included an extended summary of his achievements.

Answer:The research purpose of this paper is to use the Internet of Things technology to achieve smart agriculture under the background of big data, thus promoting the rapid development of rural revitalization.

Given the number of such errors caught, the entire text should be rewritten and checked for editorial correctness again.

Answer:The text was rewritten and checked for correctness.

Also, its correctness in terms of English language is questionable to me, so it should be checked by a native-speaker who is familiar with this field of agricultural technology, information technology and knows the relevant technical vocabulary.

Answer:To find a professional for inspection

Round 2

Reviewer 2 Report

Only some indicated editorial errors have been corrected in the article. No indicated errors or factual deficiencies have been corrected. The comments in the review have not been addressed. As such, the article does not contribute new content and cannot be published.

Author Response

Comments and Suggestions for Authors

Only some indicated editorial errors have been corrected in the article. No indicated errors or factual deficiencies have been corrected. The comments in the review have not been addressed. As such, the article does not contribute new content and cannot be published.

Answer: This experiment was conducted in the laboratory of China Agricultural University.

Agricultural greenhouses, as one of the most effective planting methods in agricultural planting, can be used as the research site to avoid the influence of other factors and make the experimental data more accurate.

Based on the analysis of previous research results and comparison with my own research results, this paper finds that under the background of big data, not only the smart agriculture based on the Internet of Things and rural revitalization and development have been greatly improved, but also promote the progress of urban and rural integration

Round 3

Reviewer 2 Report

The author has still not made the important, substantive changes and corrections to the text indicated earlier. The research methodology is not stated, the purpose and research hypotheses are not specified there is no discussion of the results, no description of the research instruments.

The legibility of the charts has not been improved.

An article cannot be published without being amended.

Author Response

Comments and Suggestions for Authors

The author has still not made the important, substantive changes and corrections to the text indicated earlier. The research methodology is not stated, the purpose and research hypotheses are not specified there is no discussion of the results, no description of the research instruments.

Answer: I added the purpose of the article, introduced the research methods of the article, and discussed the summary and the end of the article. As for the experimental instruments, I will briefly introduce their uses.

The legibility of the charts has not been improved.

Answer: I have supplemented Table 3, Table 4, Figure 5 and Figure 6.An article cannot be published without being amended.
